# Optimization of the Green Chemistry-like Extraction of Phenolic Compounds from Grape (*Vitis labrusca* L.) and Blackberry (*Rubus fruticosus* L.) Seeds with Concomitant Biological and Antioxidant Activity Assessments

**DOI:** 10.3390/plants12142618

**Published:** 2023-07-11

**Authors:** Tufy Kabbas Junior, Cristiane de Moura, Thiago Mendanha Cruz, Mariza Boscacci Marques, Mariana Araújo Vieira do Carmo, Carolina Turnes Pasini Deolindo, Heitor Daguer, Luciana Azevedo, Daniel Granato

**Affiliations:** 1Department of Chemistry, State University of Ponta Grossa (UEPG), Av. Carlos Cavalcanti, 4748, Ponta Grossa 84030-900, Parana, Brazil; tufy_kj@hotmail.com (T.K.J.); tiane.moura@hotmail.com (C.d.M.); mcruz.thiago01@gmail.com (T.M.C.); marizaboscacci@yahoo.com.br (M.B.M.); 2Nutrition Faculty, Federal University of Alfenas, Rua Gabriel Monteiro da Silva, 714, Alfenas 37130-000, Minas Gerais, Brazil; marianavieira06@hotmail.com (M.A.V.d.C.); lucianaazevedo2010@gmail.com (L.A.); 3Brazilian Ministry of Agriculture, Livestock, and Food Supply (MAPA), Federal Agricultural Defense Laboratory, São José 88102-600, Santa Catarina, Brazil; carolinaturnes.pd@gmail.com (C.T.P.D.); heitor.daguer@agricultura.gov.br (H.D.); 4Bioactivity and Applications Laboratory, Department of Biological Sciences, Faculty of Science and Engineering, School of Natural Sciences, University of Limerick, V94 T9PX Limerick, Ireland

**Keywords:** grape, blackberry, seeds, phenolic compounds, bioactivity, functional food

## Abstract

The objective of this work was to determine the phenolic composition, chemical and cellular antioxidant activity, cytotoxicity in human cells, and peroxidative inhibition of the defatted fraction of grape (*Vitis labrusca*) and blackberry (*Rubus fruticosus*) seeds. Soxhlet extraction (Sox) was used to extract the fat and obtain the degreased material. A statistical optimization study was developed to maximize the extraction of bioactive compounds and antioxidant activity from defatted grape and blackberry seeds. Simultaneous optimization was applied with a combination of 35.9 min of extraction and a solid-to-solvent ratio of 1 g of defatted grape seed to 61.28 mL of an extracting solvent (60% ethanol) and 62.1 min of extraction and a solid-to-solvent ratio of 1 g of defatted blackberry seed to 64.1 mL of an extracting solvent (60% ethanol). In the cell viability assay, HepG2 cancer cells seemed more sensitive to grape and blackberry extracts, while Ea.hy926 hybrid cells showed more resistance to their effects. In general, the extracts presented low/no cytotoxicity, exhibited a protective effect against H_2_O_2_-induced ROS production, and demonstrated antioxidant activity and a protective effect on the erythrocytes when subjected to hypotonic and isotonic conditions not presenting hemolytic behavior (5.0 to 10.0 μg GAE/mL). Thus, the results provided a broad assessment of the bioactivity of the extracts obtained using a simple and low-cost process developed by employing non-toxic solvents and with the potential to be used in technological applications.

## 1. Introduction

Agro-industrial activity generates millions of tons of by-products and plant residues, which could be repurposed in the production chain to reduce their environmental impact and gain additional value in the processes [1]. This is achieved by employing the biorefinery concept, where each component of the production chain is considered a potential product. In addition, the bioactive agents found in the biomass have many potential applications as nutraceuticals due to the possibility of reducing the risk of diseases [2].

In wine production, large amounts of waste are generated, around 20% of the total weight (bagasse—skin, seed, etc.), which is increasingly used to recover essential oils, phenolic compounds, and fibers, becoming a sustainable waste management strategy. In addition, their reuse avoids problems arising from this waste disposal, which might promote a circular bioeconomy [3,4,5,6]. Studies carried out by Gómez-Mejía et al. [7] demonstrated that grape seeds, after extraction of the oil fraction, contain catechin, gallic acid, and quercetin, in addition to presenting antioxidant activity, which has led to their use for technological purposes. However, many studies have investigated potentially toxic organic solvents such as methanol and HCl used in such processes [8]. For this reason, generating technology for producing these extracts, such as green chemistry methods, becomes vital to produce grape seed extracts with potential applications in the food and pharmaceutical industries.

Blackberry seeds are considered agricultural waste since they are usually discarded in processing industries (20% in weight). Pap et al. [9] reviewed various types of berries, including blackberries, and showed that the aqueous extract of blackberries (the pulp plus the seeds) showed antioxidant, hypertensive, antihyperglycemic, and anti-inflammatory in vitro activity. Junior et al. [10] studied blackberry seeds’ non-polar extract (hexane). They observed cytotoxic and antimicrobial activity, even suggesting using blackberry seeds as a source of antioxidant lipids. Few studies on blackberry seeds were reported, and even fewer on defatted seeds, highlighting the gap in the literature and the need to study natural resources.

Flavonoids, such as flavanols and flavonols, stilbenes, such as *trans*-resveratrol and phenolic acids, such as gallic acid, and vanillic, syringic, and caffeic acids, form the basis of the phenolic composition of grapes and their derivatives, as shown in Figure 1 [11]. In studies carried out by Fiume et al. [12], extractable phenolics in grapes are arranged as follows: about 10% are present in the pulp, 28–35% in the skin, and 60–70% in the seeds.

Antioxidant, anti-inflammatory, anticancer, neuroprotective, and antihypertensive effects, among other functions, are some of the various biological activities, both in vitro and in vivo, determined in previous studies [14,15,16,17,18]. Studies performed by Hakimuddin et al. [19], Singh et al. [20], Schlachterman et al. [21], and Singletary et al. [22] showed anticancer and antimetastatic activity when tested in vivo and in vitro against cancer cells using specific phenolic compounds (anthocyanidins, proanthocyanidins, and resveratrol) extracted from wines and seeds.

A broad spectrum of phytochemical compounds has been reported in blackberries, including phenolic acids, flavonoids, carotenoids, and organic acids, as illustrated in Figure 2 [23,24]. The anthocyanin subgroup of flavonoids is among the main components in blackberries [25,26]. The main anthocyanins reported in blackberries consist of cyanidin-based anthocyanins, including cyanidin-3-glucoside, followed by cyanidin-3-xyloside, cyanidin-3-malonylglucoside, cyanidin-3-dioxalylglucoside, and cyanidin-3-sambubioside [27].

Biological activities related to blackberries have been described in different studies, such as their protective effects against oxidative stress, endotoxicity, age-related neurodegenerative diseases, obesity, cancer, and cardiovascular diseases [28,29,30].

As important as the chemical composition and reuse of blackberry and grape by-products is, the use of non-toxic solvents in the extraction of bioactive compounds from this type of waste is for the possible use of their extracts in the food industry. The use of solvents approved by regulatory agencies, such as the EFSA (European Food Safety Authority), is highly relevant to obtain extracts that can be used to produce food with bioactive properties. Deolindo et al. [31] studied the extract of Bordeaux grape seeds (60% ethanol) and used it to produce Petit Suisse cheese, a source of phenolic compounds. Karnopp et al. [32] investigated the extract of Bordeaux grape seed skin (ethanol 70%) to prepare functional yogurt with antioxidant activity.

In an attempt to contribute to the knowledge about the association between the chemical composition and the biological activity of extracts obtained from agro-industrial residues, this study aims to optimize a system to extract bioactive compounds from defatted grape (*Vitis labrusca*) and blackberry seeds (*Rubus fruticosus)* and evaluate their bioactivity in chemical and biological systems in vitro. Notably, the study of oil (degreasing) is described by Junior et al. [10].

## 2. Results and Discussion

### 2.1. Effect of Extraction Time and Solid-to-Solvent Ratio on the Total Phenolic Composition and Antioxidant Activity

When the mean values of the total phenolic composition (TPC), antioxidant activity, flavonoids, tannins, and *ortho*-diphenols in grape (Table 1) and blackberry (Table 2) were analyzed, a significant difference was observed in all the responses (*p* < 0.05). The TPC content ranged between 2747 and 3623 mg of gallic acid equivalent per 100 g of the sample (GAE/100 g) in defatted grape seeds and between 923 and 1243 mg GAE/100 g in defatted blackberry seeds. The highest (*p* < 0.05) TPC content was obtained with a 35.9 min extraction time and 1:50 *w*/*v* solid-to-solvent ratio with grape seeds, not presenting a significant difference in relation to the extraction in 40 min with a 1:60 *w*/*v* solid-to-solvent ratio or 60 min with a 1:60 *w*/*v* solid-to-solvent ratio with defatted blackberry seeds.

Table 3 shows that the multiple regression model (e.g., MRM, Table 4) for the TPC explained 89% of the grape data variability (R^2^_adj_ = 0.888) and 66% of the blackberry data variability (R^2^_adj_ = 0.664), revealing that the TPC (grape) mathematical model can be used for predictive purposes.

The antioxidant activity in relation to the 2,2-diphenyl-1-picrylhydrazyl (DPPH) radical and ferric-reducing antioxidant power (FRAP) values of defatted grape seeds ranged between 5676 and 7069 mg of ascorbic acid equivalent per 100 g of the sample (AAE/100 g) and from 7294 to 10,023 mg AAE/100 g, respectively. For both responses, the extraction performed with a time of 40 min and a 1:60 *w*/*v* solid-to-solvent ratio showed higher mean values (*p* < 0.05), with no significant difference in relation to the extraction performed with a time of 35.9 min and a 1:50 *w*/*v* ratio. The models explain 68% and 76% of the data variability, respectively, for DPPH and FRAP (R^2^_adj_ = 0.680 and R^2^_adj_ = 0.765). Deolindo et al. [31] studied Bordeaux grape seeds and reported DPPH and FRAP values of 12,872 mg AAE/100 g and 7733 mg AAE/100 g, respectively, using a 1:50 *w*/*v* ratio (60% ethanol) and obtaining a FRAP value close to that found in this study.

The values obtained for defatted blackberry seeds regarding Fe^2+^ and FRAP chelation ranged from 144 to 235 mg of ethylenediamine tetraacetic acid equivalent per 100 g of the sample (EDTA/100 g) and 3060 to 4295 mg AAE/100 g, respectively. The extraction that presented the highest mean values (*p* < 0.05) was performed for 50 min with a 1:64.1 *w*/*v* solid-to-solvent ratio for both responses. The models explained 96% and 68% of the data variability for Fe^2+^ and FRAP (R^2^_adj_ = 0.963 and R^2^_adj_ = 0.679), respectively. Junior et al. [10], studying non-polar blackberry seed extract, reported Fe^2+^ values of 261 mg EDTA/100 g, which is also close to the value found in this study.

It is relevant to emphasize the importance of evaluating the antioxidant activity of natural products using different methods since bioactive compounds can display one or more mechanisms of action, such as hydrogen atom transfer, single electron transfer, and the capacity to chelate transition metals [33]. Phenolic compounds are recognized as potential antioxidants because they capture ROS, reducing and chelating ions and thus decreasing lipidic peroxidation in vivo.

This study used simultaneous optimization based on the desirability function to determine the best experimental conditions to obtain a high phenolic content and significant antioxidant activity (Appendix B Figure A1). With a 0.899 desirability function (*d*-value) for grape seeds and 0.892 for blackberry seeds, the following experimental conditions were proposed: 35.9 min of extraction and the solid-to-solvent ratio of 1 g defatted grape seed to 61.28 mL extractor solvent and 62.1 min of extraction and the solid-to-solvent ratio of 1 g defatted blackberry seed to 64.1 mL extractor solvent. The values predicted by the regression models, experimental values, and absolute error percentage are presented below (Table 5). The experimental data are within this prediction band, confirming the suitability of the models since they could predict the phenolic composition and antioxidant activity of the ethanolic extract of defatted grape and blackberry seeds.

### 2.2. Chemical Composition of the Optimized and Lyophilized Extracts

The chemical composition and antioxidant activity values of the optimized extracts are shown in Table 6. The contents of flavonoids, condensed tannins, and *ortho*-diphenols in defatted grape seeds were 61 ± 0.05 mg QE/100 g, 2609 ± 257 mg CTE/100 g, and 729 ± 81 mg ACE/100 g, respectively. As regards blackberry seeds, the values found were 38.1 ± 0.06 mg QE/100 g (flavonoids), 409 ± 154 mg CTE/100 g (tannins), and 233 ± 37 mg ACE/100 g (*ortho*-diphenols). Deolindo et al. [31] studied grape seed hydroalcoholic extract (60% ethanol) and obtained higher values of these compounds, which reached 6736 ± 276 mg QE/100 g (flavonoids), 10,635 ± 753 mg CTE/100 g (tannins), and 1260 ± 9 mg ACE/100 g (*ortho*-diphenols), respectively.

### 2.3. Phenolic Composition by LC-MS

The compounds identified in the grape- and blackberry-optimized extracts are presented in Table 7. These include 3,4-dihydroxybenzoic acid (5.7 ± 0.1 mg/100 g), catechin (573.8 ± 11.2 mg/100 g), *p*-coumaric acid (1.2 ± 0.0 mg/100 g), and quercetin (231.7 ± 1.2 mg/100 g), identified in the grape extract. The same compounds were reported in the analyses carried out by Gómez-Mejía et al. [7], 147 ± 5, 295 ± 37, 7.8 ± 0.9, and 112 ± 5 µg/g, respectively. When the blackberry extracts were analyzed, the compounds identified were 3,4-Dihydroxybenzoic acid, catechin, *p*-coumaric acid, ferulic acid, rutin, epicatechin, and caffeic acid. These were all reported in the study developed by Halin et al. [34], which employed extract (ethanol 80%) taken from various parts of the blackberry.

### 2.4. Antihemolytic Activity

The in vitro antihemolytic activity of the extracts (60 µg/mL) was evaluated under different osmolarity conditions ([NaCl] = 0.1 to 0.9%) (Figure 3). We observed that the extracts either presented a lower percentage of hemolysis or did not differ from the control (the absence of the extract) in all NaCl concentrations, demonstrating a protective effect on the erythrocytes when subjected to hypotonic and isotonic conditions. Moreover, they did not present hemolytic behavior in relation to the tested concentration. The decrease in the concentration of NaCl in the solution causes water absorption by the erythrocyte, making the cells swell and, consequently, provoking leakage of the intracellular content. The presence of phenolic compounds helps reduce the erythrocyte membrane’s fluidity through hydrogen bonds between the membrane phospholipids (polar heads) and phenolic compounds [35,36].

The NaCl concentration needed for the hemolysis of 50% of the erythrocytes (H_50_) is a way of evaluating the ability of the extract to protect the membrane from hemolysis. In this study, the optimized grape extract (60 µg/mL) obtained 0.437% H_50_, while the control (the absence of the extract) obtained 0.446% H_50_. The optimized blackberry extract (60 µg/mL), in turn, obtained 0.439% H_50_, while the control (the absence of the extract) obtained 0.446% H50. Atrooz, Harb, and Al-Qato [37] investigated the ethanolic extract of various seeds, and their results presented an antihemolytic effect, pointing out that the grape extract obtained better results. Our results reinforce the absence of toxicity in vitro in the extract of these seeds on erythrocytes, despite not presenting a significant protective effect.

### 2.5. Intracellular ROS Generation and Cell Viability

Bioactive compounds found in berries, such as phenolic compounds, flavonoids, and tannins, can individually or synergistically protect individuals against several disorders due to their antioxidant properties. Such protective activity includes free radical scavenging, protection from DNA damage, induction of apoptosis, and inhibition of the growth and proliferation of cancer cells, just to name a few [38]. In this study, neither grape nor blackberry extracts induced ROS generation in A549 and HCT8 cancer cells, and they even exhibited a protective effect against H_2_O_2_-induced ROS production (Figure 4). This antioxidant activity may affect the pro-tumorigenic cancer cells signaling, interfering with their viability (A549 grape: IC_50_ = 900.9 µg/mL; blackberry: IC_50_ = 753.1 µg/mL; HCT8 grape: IC_50_ = 423.0 µg/mL) and proliferation (A549 grape: GI_50_ = 548.2 µg/mL; blackberry: GI_50_ = 100.2 µg/mL; HCT8 grape: GI_50_ = 322.1 µg/mL; blackberry: GI_50_ = 714.3 µg/mL), as observed in this study. In a different way, both grape and blackberry induced oxidative stress in HepG2 cancer cells, which can explain the cytotoxicity observed in the cell viability assay (grape: IC_50_ = 229.0 µg/mL; GI_50_ = 95.6 µg/mL; blackberry: GI_50_ = 424.5 µg/mL) (Figure 5). The increase in ROS at toxic levels can activate the JNK pathway, leading to cell death [39,40]. However, in HepG2, the extracts reduced the oxidative stress induced by H_2_O_2_, presenting antioxidant activity. The EA.hy926 cell line seemed to be more resistant to the effects of the extracts once it exhibited higher values for all the parameters (blackberry: IC_50_, GI_50_, and LC_50_ > 1000.0 µg/mL; grape: IC_50_ = 968.4 µg/mL, GI_50_ = 746.4 µg/mL and LC_50_ > 1000.0 µg/mL) in comparison with cancer cells (HCT8, HepG2, and A549). Non-cancer cells present specific adaptations to overcome the damaging effects of ROS through the balanced generation of these species, sufficient antioxidant activity, and cellular repair, which leads to low amounts of ROS toward limited cell survival and proliferation [39]. Regarding EA.hy926 hybrid cells, both extracts protected them against peroxide effects, reducing the ROS levels below the baseline, and in general, this effect was also observed in this study, considering the low or no toxicity in the cell viability test. The grape and blackberry antioxidant capacity can be correlated with all antioxidant in vitro assays (FRAP, DPPH, and the TPC), especially their high ability to scavenge hydroxyl radicals.

## 3. Material and Methods

### 3.1. Chemical Reagents and Cells

The Folin-Ciocalteu reagent, isobutanol, quercetin (95% purity), sodium hydroxide (NaOH), 2,2-diphenyl-1-picryl-hydrazyl radical (DPPH), pyrocatechol violet (3,3′ acid, 4-trihydroxyfuchsone-2′-sulfonic acid), 2,4,6-tris (2-pyridyl)-S-triazine (TPTZ), hydrogen peroxide, heneicosanoic acid (C:21), cholesterol, dimethyl sulfoxide (DMSO), 2′,7′-dichlorofluorescein diacetate (DCFH-DA), dehydrated ferric chloride, ascorbic acid, and ferrozine 3-(2*-*Pyridyl)-5*,*6*-*di*(*2*-*furyl)-1*,*2*,*4*-*triazine*-*5*′,*5*″-*disulfonic acid disodium salt) were obtained from Sigma-Aldrich (São Paulo, Brazil). Anhydrous sodium sulfate, *n*-hexane, chloroform, and methyl alcohol were obtained from Anidrol (São Paulo, Brazil). Ethyl alcohol was obtained from Neon (São Paulo, Brazil). A549, IMR90, HepG2, and Caco-2 were obtained from the cell bank of Rio de Janeiro/Rio de Janeiro, Brazil.

### 3.2. Vegetable Material

Blackberry (*Rubus fruticosus)* and Bordeaux grape *(Vitis labrusca* cv. Bordeaux) were produced in Vacaria, Rio Grande do sul (29°32′30″ S and 50°54′51″ W) and Garibaldi, Rio Grande do Sul (29°15′22″ S and 51°32′01″ W). The fruit, leaves, and flowers were identified morphologically by Prof. Rosangela Capuano Tardivo, and exsiccates of the vegetable material were deposited at the herbarium of the State University of Ponta Grossa and registered with numbers 22494 and 22495, respectively. The seeds were obtained manually and were subjected to sanitization with 100 mg/L NaOCl for 15 min, washing (current water), and drying (air circulation oven (Tecnical, Modelo TE-393/1, São Paulo, Brazil)) at 35 °C for 48 h and then milled (analytical mill (QUIMIS-6298A21)), standardized (Tyler sieve, mesh 60), and stored under refrigeration.

### 3.3. Obtaining Hydroalcoholic Extracts and Extraction Optimization

To degrease the seeds, 5.0 g of ground seeds were weighed and transferred to the extraction cartridge. The extractions were carried out for 5 h in a Soxhlet apparatus with *n*-hexane as the extraction solvent. The temperature was kept constant in the extractor apparatus, within the boiling range of *n*-hexane (68–70 °C). The degreased material was removed from the cartridge and dried in an oven at 35 °C for 2 h and stored in a refrigerator (5 ºC) until the extraction of the phenolic fraction.

A two-factor rotational central composite design was used for each vegetable material to verify the influence of the extraction time and solid-to-solution ratio on the total phenolic content and chemical antioxidant activity. To achieve this aim, 11 assays were randomly tested (Appendix C Table A1), with minimum extraction time values of 35.9 min and a maximum of 64.1 min, and 1:35.9 g/mL minimum and 1:64.1 g/mL maximum solid-to-solution ratios. The multiple linear regression methodology was used following the recommendations and procedures described by Granato et al. [41].

The proposed mathematical models only included regression coefficients with significance (*p* < 0.05), and the quality of the models was evaluated by the coefficient of determination (R^2^), an adjusted R^2^, and the residue analyzed using the Shapiro–Wilk test. Using RSM, regression models were obtained to describe each functional property and chemical compound extracted as a function of time and the sample/solvent ratio. Data were presented as the mean ± combined standard deviation. To compare the response variables, the data normality and homoscedasticity were initially checked using the Shapiro–Wilk and Brown–Forsythe tests, respectively, and the differences between the mean values were evaluated using one-way variance analysis (ANOVA) followed by the Fisher test. *p*-values below 5% were considered for rejecting the null hypothesis. TIBCO Statistica 13.3 software (TIBCO Statistica Ltd., Palo Alto, CA, USA) was used for all statistical analyses.

### 3.4. Phenolic Composition by LC-MS/MS and UV-VIS Spectrophotometry

The TPC was assessed using the Prussian Blue assay [42], and the results were expressed in mg of gallic acid equivalent per 100 g of sample (mg GAE/100 g). The total flavonoid content (TFC) of the samples was determined using the aluminum chloride colorimetric assay described by Aguiar et al. [43], and the results were expressed as mg of catechin equivalent per 100 g (mg CE/100 g) of the sample. The total *ortho*-diphenolic content (TOC) was determined using a sodium molybdate assay [44], and the results were expressed as mg of chlorogenic acid equivalent per 100 g (mg CAE/100 g). The total condensed tannin content (TCT) was determined using the vanillin-H_2_SO_4_ assay, and the results were expressed as (mg CTE/100 g).

In total, 22 phenolic compounds were individually determined by liquid chromatography coupled with tandem mass spectrometry (LC-MS/MS), according to the method described by Seraglio et al. [45]. LC-MS/MS analysis was performed with solvents of chromatographic grade and ultrapure water (minimum resistivity 18.3 MΩ cm). The analyses were performed in an ExionLC chromatograph coupled with a QTRAP 5500 mass spectrometer, both supplied by AB Sciex LLC (Framingham, MA, USA). The LC-MS/MS equipment was used in electrospray ionization source and multiple reaction monitoring (MRM) modes. Analyst software 1.6.2 (AB Sciex, Foster City, CA, USA) and MultiQuant (AB Sciex, Foster City, CA, USA) were used for instrument operation and data processing, respectively.

The chromatographic separation was performed in reversed-phase mode with a Zorbax Eclipse Plus C_18_ column (3.5 µm, 3.0 × 100 mm) manufactured by Agilent Technologies, Inc. (Santa Clara, CA, USA). The column oven was kept at 40 °C; the sample injection volume was 5 μL, and the mobile phase flow rate was set to 300 μL/min. Liquid chromatography was performed in gradient elution, as follows: 98% A (0–3 min), 80% A (3–10 min), 10% A (10–11 min), and 98% A (11–13 min), plus 4 min for the system to reach equilibrium. Mobile phase A comprised a formic acid aqueous solution (0.1%), and mobile phase B was composed of acetonitrile with 0.1% formic acid. Each sample was injected in triplicate, and the quantitation of each phenolic compound was performed by external standard calibration in the 25–1000 μg/L linear range, except for EGCG, whose linear range was 500–1000 μg/L, and correlation coefficients above 0.9980. The limits of quantitation ranged from 0.20 to 12.8 μg/L, and the description of the method validation is already available in Seraglio et al. [45]. The positive and negative ionization modes were set to 5500 V and −4500 V, respectively; other instrument parameters were set to 25 psi (curtain gas) and 55 psi (nebulizer gas and auxiliary gas) at 400 °C in a nitrogen atmosphere. The mass spectrometry parameters for the analysis of each phenolic compound are shown in Appendix A, Appendix A. The results were expressed in the mg of each phenolic compound per 100 g of extract (mg/100 g).

### 3.5. Antioxidant Activity

To evaluate antioxidant activity, tests assessing distinct action mechanisms were used, and the readings were made using a microplate reader (Synergy™ H1, Biotek, Shoreline, WA, USA), according to the methods proposed by Santos, Brizola, and Granato [46]. The antioxidant activity was evaluated by the Fe^2+^ chelating capacity using ferrozine as the chromophore, and the results were expressed in mg of the EDTA equivalent (mg EDTAE/100 g sample), DPPH free radical scavenging activity, using a methanolic solution of DPPH at 0.10 mmol/L, and FRAP. For DPPH and FPRA assays, the results were expressed in mg of the ascorbic acid equivalent (mg AAE/100 g sample). To assess the inhibition of induced oxidation, egg yolk was used as the source of phospholipids and triacylglycerols [47]. The analysis was conducted at a pH of 7.4, and quercetin was used as the standard. The results were expressed in mg of QE/100 g. The ability to capture ^•^OH radicals was determined by UV-VIS spectrophotometry in the presence of salicylic acid [48]. The hydroxyl radical inhibition rate was calculated using the following equation:% Inhibition = ((A_Sample_ − A_Blank_)/(A_Control_ − A_Blank_)) × 100(1)

### 3.6. Toxicity Profiling and Cellular Antioxidant Activity

To assess the in vitro cytotoxicity of the extract, tetrazolium salt staining was used to evaluate the extract’s effect on cell viability by the parameters IC_50_, GI_50_, and LC_50_, as previously described by Carmo et al. [49]. A549 (lung adenocarcinoma epithelial cells), HepG2 (human hepatoma carcinoma cells), HCT8 (human colon carcinoma), and EA.hy926 (endothelial cell hybrid) were seeded at a density of 1 × 10^4^ cells/well. They were treated with extracts for 48 h, using the following concentrations: 50, 100, 150, 200, 250, and 500 µg of QE/mL. Then, an MTT solution (5 mg/mL) was added, and after 4 h, the insoluble formazan produced was dissolved in 100 μL of DMSO. Absorbance (570 nm) was measured using a microplate reader, and the dose–response analysis was determined by non-linear regression (curve fit).

Reactive oxygen species (ROS) generation impacts oxidation in biological systems. The intracellular amounts of ROS were measured by a fluorometric assay with 2,7-dichlorofluorescein diacetate (DCFH-DA), which detects and quantifies intracellular production of the superoxide radical, hydroxyl radical, and hydrogen peroxide [50]. All cell lines were seeded at 6 × 10^4^/well and treated for 1 h at 37 °C with extracts diluted in a DCFH-DA solution (25 mmol/L) at 10, 50, and 100 µg QE/mL concentrations. The cells were treated with 15 μmol/L of H_2_O_2_ for the positive control and for the negative control, only with the culture medium. After incubation, the intracellular production of ROS was measured by fluorometric detection of DCF oxidation using a spectrofluorometer with an excitation wavelength of 485 nm and an emission wavelength of 538 nm [51]. The DCF fluorescence intensity was proportional to the amount of ROS formed intracellularly.

### 3.7. In Vitro Antihemolytic Effect

Under hypotonic conditions, the antihemolytic activity was evaluated with erythrocytes isolated from O^+^ blood samples obtained from the Wallace Thadeu de Mello e Silva Regional University Hospital. The assays were performed according to Migliorini et al. [52], with hematocrit 0.8% and [NaCl] = 0.1%, 0.4%, and 0.8% (*w*/*v*), and the extract concentrated to 5.0, 7.5, and 10.0 μg GAE/mL, and the hemolysis rate was measured by the absorbance at 540 nm. The hemoglobin oxidation was carried out by the absorbance measure at 630 nm [53]. The entire experimental procedure was previously approved by the State University of Ponta Grossa Ethics Committee (CAAE 94830318.1.0000.0105).

### 3.8. Statistical Analysis

The experimental data were presented as the means ± sample standard deviation. When appropriate, a comparison of means between groups was performed by one-way variance analysis (ANOVA), followed by Fisher’s multiple comparison test. Before that, the Brown–Forsythe test was employed to verify the homoscedasticity of the entire data set using TIBCO Statistica 13.3 software (TIBCO Statistica Ltd. Pao Alto, CA, EUA) [10].

## 4. Conclusions

A simultaneous optimization was applied, and a combination of 35.9 min of extraction and a solid-to-solvent ratio of 1 g defatted grape seed to 61.28 mL extracting solvent (60% ethanol) and 62.1 min of extraction and a solid-to-solvent ratio of 1 g defatted blackberry seed to 64.1 mL of a 60% ethanol solution were the combinations that obtained the best results. Grape and blackberry seed extracts did not induce ROS generation in A549 and HCT8 cancer cells and, in addition, exhibited a protective effect against H_2_O_2_-induced ROS production and demonstrated antioxidant activity and a protective effect on erythrocytes when subjected to hypotonic and isotonic conditions. Furthermore, the extracts did not present hemolytic behavior in relation to the tested concentration. Considering the requirements of the 2030 Agenda and the need for sustainable ways to produce food ingredients, our results provide a broad assessment of the bioactivity of extracts obtained employing a simple and low-cost process using non-toxic solvents, with potential use in technological applications.

## Figures and Tables

**Figure 1 plants-12-02618-f001:**
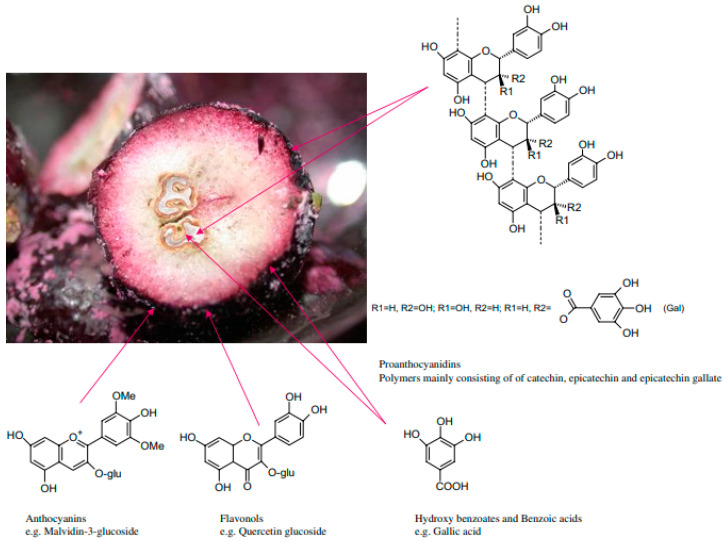
Main phenolic compounds in different grape fractions [13].

**Figure 2 plants-12-02618-f002:**
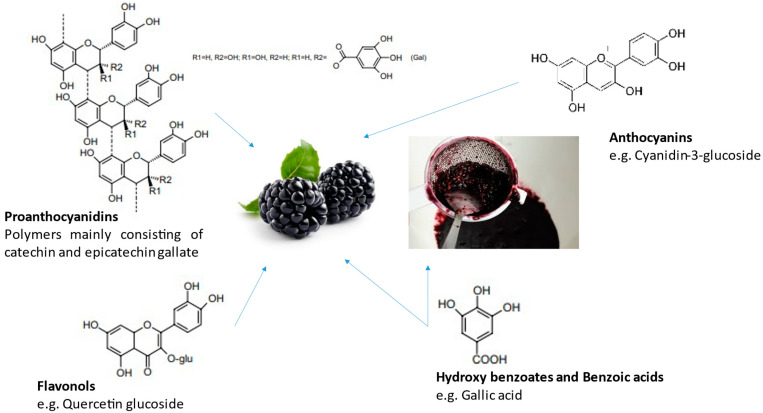
Main phenolic compounds present in different blackberry fractions.

**Figure 3 plants-12-02618-f003:**
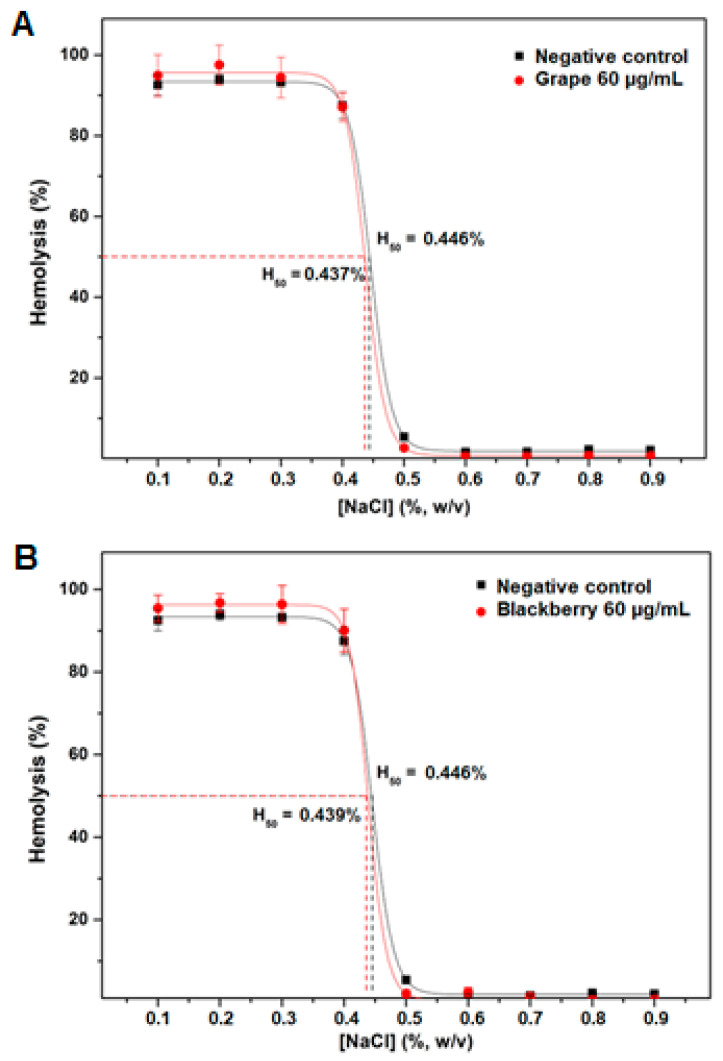
Effect of lyophilized grape (**A**) and blackberry (**B**) extracts on the hemolysis of human erythrocytes in the hypotonic medium.

**Figure 4 plants-12-02618-f004:**
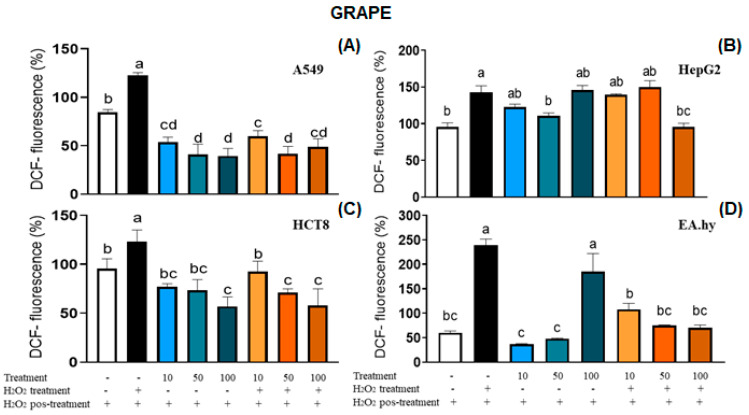
Results of intracellular measurement of ROS in A549 (**A**), HepG2 (**B**), HCT8 (**C**), and EA.hy (**D**) cells by spectrofluorometry. Optimized treatment and lyophilized extract of grape seeds (*Vitis labrusca* cv. Bordeaux) and blackberry seeds (*Rubus fruticosus)* at 10–100 µg/mL. Quantitative data are the mean ± standard deviation. Different letters represent statistically significant differences (*p* ≤ 0.05).

**Figure 5 plants-12-02618-f005:**
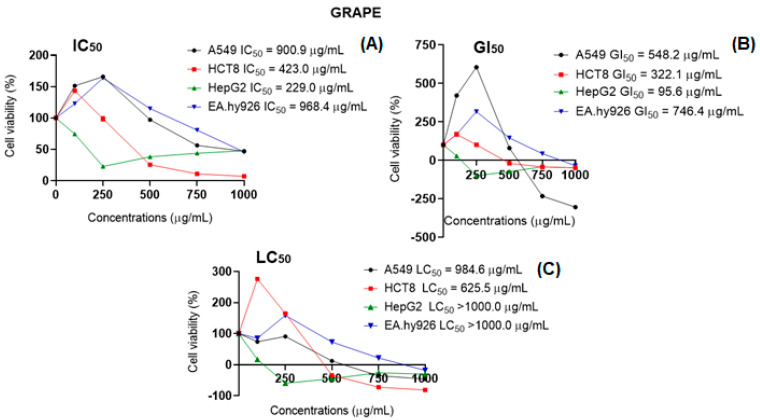
Cell viability and evaluation of the concentration-dependent effect after 48-h exposure to lyophilized hydroalcoholic extract in A549, HCT8, HepG2, and Ea.hy926 cell lines. (**A**) IC_50_: concentration of the agent that inhibits cell growth by 50%; (**B**) GI_50_: concentration of the agent that inhibits growth by 50%, related to untreated cells; (**C**) LC_50_: concentration of the agent that results in a net loss of 50% of cells, associated with the number of cells at the start of treatment.

**Table 1 plants-12-02618-t001:** Chemical composition and antioxidant activity of grape extracts in relation to the extraction time and solid-to-volume ratio.

Tests	Independent Variables (Original and Coded Values)		Total Phenolic Content (mg GAE/100 g)	Flavonoids (mg QE/100 g)	Condensed Tannins (mg CTE/100 g)	*Ortho*-Diphenolics (mg [ACE]/100 g)	DPPH (mg AAE/100 g)	Fe^2+^ Chelating Capacity (mg EDTAE/100 g)	FRAP (mg AAE/100 g)
	Time (min)	1:X (g/mL)							
1	40 (−1)	1:40 (−1)	2916 ± 7 ^cde^	62.9 ± 1.7 ^f^	3242 ± 95 ^e^	554 ± 20 ^e^	5870 ± 98 ^def^	66.1 ± 1.8 ^c^	9032 ± 328 ^b^
2	40 (−1)	1:60 (1)	3609 ± 24 ^a^	78.8 ± 2.8 ^b^	3928 ± 315 ^de^	744 ± 24 ^bc^	7069 ± 544 ^a^	55.5 ± 4.9 ^d^	10,023 ± 98 ^a^
3	60 (1)	1:40 (−1)	2914 ± 61 ^de^	55.9 ± 2.1 ^g^	3885 ± 353 ^de^	697 ± 40 ^cd^	5676 ± 250 ^f^	45.4 ± 4.1 ^e^	7518 ± 241 ^de^
4	60 (1)	1:60 (1)	2747 ± 117 ^f^	74,1 ± 1.6 ^c^	4982 ± 438 ^bc^	700 ± 78 ^cd^	6173 ± 36 ^cde^	85.3 ± 5.3 ^a^	7294 ± 234 ^e^
5	35.9 (−1.41)	1:50 (0)	3623 ± 105 ^a^	72.7 ± 3.0 ^cde^	5561 ± 1092 ^b^	963 ± 64 ^a^	6650 ± 141 ^ab^	46.7 ± 2.6 ^e^	9596 ± 308 ^a^
6	64.1 (1.41)	1:50 (0)	3036 ± 45 ^bc^	69.4 ± 2.6 ^de^	5030 ± 538 ^bc^	801 ± 54 ^b^	5832 ± 79 ^def^	65.1 ± 6.5 ^c^	8360 ± 190 ^c^
7	50 (0)	1:35.9 (−1.41)	2825 ± 12 ^ef^	68.5 ± 3.3 ^e^	3993 ± 449 ^de^	892 ± 11 ^a^	5517 ± 80 ^f^	36.1 ± 2.4 ^f^	7807 ± 166 ^d^
8	50 (0)	1:64.1 (1.41)	3010 ± 51 ^bcd^	73.6 ± 3.9 ^cd^	5695 ± 267 ^ab^	753 ± 43 ^bc^	6679 ± 511 ^ab^	74.2 ± 3.7 ^b^	7565 ± 579 ^de^
9C	50 (0)	1:50 (0)	3087 ± 72 ^b^	70.1 ± 2.7 ^cde^	6384 ± 368 ^a^	787 ± 80 ^b^	6388 ± 91 ^bc^	61.4 ± 1.3 ^cd^	8661 ± 342 ^bc^
10C	50 (0)	1:50 (0)	3063 ± 100 ^b^	69.2 ± 2.3 ^de^	4547 ± 411 ^cd^	636 ± 16 ^d^	5786 ± 293 ^ef^	64.2 ± 7.2 ^c^	8305 ± 202 ^c^
11C	50 (0)	1:50 (0)	2884 ± 89 ^e^	84.9 ± 3.0 ^a^	4022 ± 128 ^de^	648 ± 12 ^d^	6237 ± 153 ^bcd^	61.7 ± 2.9 ^cd^	8660 ± 249 ^bc^
	*p*-Value (homocedasticity)		0.503	0.999	0.942	0.914	0.709	0.954	0.97
	*p*-Value (one-way ANOVA)		<0.001	<0.001	<0.001	<0.001	<0.001	<0.001	<0.001

Note: (C) = central point; 1:X = 1 g vegetable material to X mL extractor solvent; CTE = catechin equivalent; GAE = gallic acid equivalent; QE = quercetin equivalent; AAE = ascorbic acid equivalent; EDTAE = ethylenediaminetetraacetic acid equivalent; ACE = chlorogenic acid equivalent. Different letters in the same column indicate significant differences at the 5% level.

**Table 2 plants-12-02618-t002:** Chemical composition and antioxidant activity of blackberry extracts in relation to the extraction time and solid-to-volume ratio.

Tests	Independent Variables (Original and Coded Values)		Total Phenolic Content (mg GAE/100 g)	Flavonoids (mg QE/100 g)	Condensed Tannins (mg CTE/100 g)	*Ortho*-Diphenolics (mg ACE/100 g)	DPPH (mg AAE/100 g)	Fe^2+^ Chelating Capacity (mg EDTAE/100 g)	FRAP (mg AAE/100 g)
	Time (min)	1:X (g/mL)							
1	40 (−1)	1:40 (−1)	923 ± 11 ^h^	46.2 ± 1.6 ^e^	579 ± 44 ^f^	253 ± 32 ^e^	1769 ± 87 ^f^	144 ± 8 ^f^	3060 ± 53 ^f^
2	40 (−1)	1:60 (1)	1167 ± 6 ^g^	58.1 ± 1.7 ^cd^	637 ± 50 ^ef^	258 ± 3 ^e^	2246 ± 197 ^cde^	227 ± 1 ^a^	3823 ± 143 ^d^
3	60 (1)	1:40 (−1)	1067 ± 25 ^cd^	55.9 ± 1.8 ^d^	594 ± 42 ^f^	350 ± 35 ^cd^	2152 ± 147 ^de^	143 ± 4 ^f^	3437 ± 92 ^e^
4	60 (1)	1:60 (1)	1243 ± 13 ^a^	65.7 ± 1.0 ^a^	1001 ± 112 ^ab^	367 ± 13 ^bcd^	1229 ± 190 ^g^	215 ± 9 ^b^	4258 ± 148 ^a^
5	35.9 (−1.41)	1:50 (0)	1143 ± 9 ^de^	60.9 ± 0.6 ^b^	856 ± 19 ^cd^	426 ± 66 ^a^	2470 ± 173 ^bc^	186 ± 7 ^c^	3893 ± 78 ^cd^
6	64.1 (1.41)	1:50 (0)	1202 ± 8 ^b^	63.8 ± 1.6 ^a^	840 ± 40 ^cd^	409 ± 31 ^ab^	2366 ± 194 ^cd^	185 ± 5 ^c^	3958 ± 93 ^bcd^
7	50 (0)	1:35.9 (−1.41)	1073 ± 26 ^g^	59.4 ± 1.6 ^bc^	750 ± 45 ^de^	420 ± 42 ^ab^	2098 ± 93 ^e^	120 ± 4 ^g^	3551 ± 92 ^e^
8	50 (0)	1:64.1 (1.41)	1170 ± 21 ^cd^	60.3 ± 0.4 ^bc^	1114 ± 141 ^a^	326 ± 18 ^d^	2760 ± 127 ^a^	235 ± 5 ^a^	4295 ± 317 ^a^
9C	50 (0)	1:50 (0)	1105 ± 20 ^f^	58.4 ± 1.4 ^c^	872 ± 112 ^bcd^	382 ± 49 ^abcd^	2264 ± 78 ^cde^	175 ± 2 ^d^	3951 ± 80 ^bcd^
10C	50 (0)	1:50 (0)	1187 ± 30 ^bc^	59.7 ± 1.1 ^bc^	872 ± 20 ^bcd^	372 ± 12 ^abcd^	2706 ± 94 ^ab^	166 ± 5 ^e^	4153 ± 166 ^ab^
11C	50 (0)	1:50 (0)	1129 ± 10 ^ef^	61.1 ± 0.9 ^b^	926 ± 96 ^bc^	391 ± 15 ^abc^	2634 ± 79 ^ab^	183 ± 3 ^cd^	4088 ± 209 ^abc^
	*p*-Value (homocedasticity)		0.949	0.976	0.889	0.699	0.948	0.79	0.879
	*p*-Value (one-way ANOVA)		<0.001	<0.001	<0.001	<0.001	<0.001	<0.001	<0.001

Note: (C) = central point; 1:X = 1 g vegetable material to X mL extractor solvent; CTE = catechin equivalent; GAE = gallic acid equivalent; QE = quercetin equivalent; AAE = ascorbic acid equivalent; EDTAE = ethylenediaminetetraacetic acid equivalent; ACE = chlorogenic acid equivalent. Different letters in the same column indicate significant differences at the 5% level.

**Table 3 plants-12-02618-t003:** Effects of time and x/s ratio on the extraction of bioactive compounds and antioxidant activity of grape (*Vitis labrusca*) seeds.

Model Components	Regression Coefficients	Standard Error	*t-*Value	*p*-Value	−95% Confidence	+95% Confidence
**Total phenolic content**						
Mean	3012	29.6	102	<0.001	2951	3073
(1) Time (min)(L)	−212	18.1	−11.7	<0.001	−249	−175
Time (min)(Q)	140	21.6	6.5	<0.001	95.7	185
(2) Ratio (x/s)(L)	99	18.1	5.4	<0.001	61.3	136
Ratio (x/s)(Q)	−67	21.6	−3.1	0.005	−112	−22.5
1 L to 2 L	−215	25.6	−8.4	<0.001	−268	−162
R^2^	0.905					
Adjusted R^2^	0.888					
*p*-value (normality of residues)	0.763					
**Free radical scavenging activity (DPPH)**						
Mean.	6171	52.1	118	<0.001	6064	6279
(1) Time (min)(L)	−281	61.2	−4.6	<0.001	−408	−155
(2) Ratio (x/s)(L)	418	61.2	6.8	<0.001	292	544
R^2^	0.706					
Adjusted R^2^	0.680					
*p*-value (normality of residues)	0.945					
**Ferric-reducing antioxidant power (FRAP)**						
Mean	8542	99.6	85.8	<0.001	8336	8747
(1) Time (min)(L)	−751	61.1	−12.3	<0.001	−877	−624
Time (min)(Q)	254	72.9	3.5	0.002	103	404
Ra (io/s)(Q)	−396	72.9	−5.4	<0.001	−546	−246
1 L to 2 L	−304	86.2	−3.5	0.002	−482	−126
R^2^	0.795					
Adjusted R^2^	0.765					
*p*-value (normality of residues)	0.401					

**Table 4 plants-12-02618-t004:** Effects of time and x/s ratio on the extraction of bioactive compounds and antioxidant activity of blackberry (*Rubus fruticosus*) seeds.

Model Components	Regression Coefficient	Standard Error	*t-*Value	*p*-Value	−95% Confidence	+95% Confidence
**Total phenolic content**						
Mean	1145	6.6	173	<0.001	1131	1159
(1) Time (min)(L)	38.0	5.6	6.8	<0.001	26.5	49.6
(2) Ratio (x/s)(L)	69.7	5.6	12.5	<0.001	58.2	81.2
Ratio (x/s)(Q)	−22.6	6.4	−3.5	0.002	−35.8	−9.5
1 L to 2 L	−17.3	7.9	−2.2	0.038	−33.6	−1.0
R^2^	0.706					
Adjusted R^2^	0.664					
*p*-value (normality of residues)	0.648					
**Fe^2+^ chelating capacity**						
Mean	177	1.6	110	<0.001	173	180
Time (min)(Q)	5.4	1.5	3.5	0.002	2.2	8.6
(2) Ratio (x/s)(L)	39.7	1.4	29.3	<0.001	36.9	42.5
R^2^	0.965					
Adjusted R^2^	0.963					
*p*-value (normality of residues)	0.077					
**Ferric-reducing antioxidant power (FRAP)**						
Mean	4065	51.6	78.8	<0.001	3959	4171
(1) Time (min)(L)	113	31.6	3.6	0.002	47.9	178
Time (min)(Q)	−140	37.7	−3.7	0.001	−218	−62.4
(2) Ratio (x/s)(L)	330	31.6	10.4	<0.001	265	395
Ratio (x/s)(Q)	−142	37.7	−3.8	<0.001	−220	−63.8
R^2^	0.719					
Adjusted R^2^	0.679					
*p*-value (normality of residues)	0.232					

**Table 5 plants-12-02618-t005:** Predicted and experimental values of the optimized extracts.

	Parameters Evaluated	Predicted Mean Values	−95% Prediction	+95% Prediction	Experimental Mean Values	Relative Error (%)
Grape	TPC (mg GAE/100 g)	3847	3625	4068	3623 ± 40	5.8
	FRAP (mg AAE/100 g)	10,084	9339	10,829	9395 ± 14	6.8
	DPPH (mg AAE/100 g)	6914	6167	7661	7102 ± 164	2.7
Blackberry	TPC (mg GAE/100 g)	1249	1184	1314	1295 ± 12	3.7
	FRAP (mg AAE/100 g)	4180	3811	4549	3804 ± 45	9.0
	Fe^2+^ (mg EDTA/100 g)	241	225	256	231 ± 6	4.1

Note: TPC = total phenolic content; GAE = gallic acid equivalent; FRAP = ferric iron reduction antioxidant power; AAE = ascorbic acid equivalent; EDTAE = ethylenodiaminetetraacetic acid equivalent; DPPH = 2.2-diphenyl-1-picrylhydrazyl.

**Table 6 plants-12-02618-t006:** Chemical composition of optimized and lyophilized grape and blackberry extracts.

	Total Phenolic Content (mg GAE/100 g)	Flavonoids (mg QE/100 g)	Condensed Tannins (mg CTE/100 g)	*Ortho*-Diphenols (mg ACE/100 g)	DPPH (mg AAE/100 g)	Fe^2+^ Chelating Capacity (mg EDTAE/100 g)	FRAP (mg AAE/100 g)	OH Radical Capture (mg AAE/100 g)	Lipidic Peroxidation Inhibition—IC_50_ (mg/L)
Grape	2778 ± 218	61 ± 0.05	2609 ± 257	729 ± 81	4838 ± 654	59 ± 2	4972 ± 164	258,165 ± 17,568	124.1
Blackberry	981 ± 185	38.1 ± 0.06	409 ± 154	233 ± 37	1377 ± 117	210 ± 8	1856 ± 164	164,279 ± 5856	30.6

Note: TPC = total phenolic content; GAE = gallic acid equivalent; CTE = catechin equivalent; FRAP = ferric iron reduction antioxidant power; AAE = ascorbic acid equivalent; EDTAE = ethylenodiaminetetraacetic acid equivalent; DPPH = 2,2-diphenyl-1-picrylhydrazyl; QE = quercetin equivalent; ACE = chlorogenic acid equivalent.

**Table 7 plants-12-02618-t007:** Phenolic composition of grape and blackberry lyophilized extracts.

	Grape (mg/100 g)	Blackberry (mg/100 g)
*Phenolic acids*		
Ferulic acid	1.3 ± 0.2	0.8 ± 0.1
2,5- Dihydroxybenzoic acid	ND	33.5 ± 1.6
3,4-Dihydroxybenzoic acid	5.7 ± 0.1	34.5 ± 0.9
Salicylic acid	ND	0.2 ± 0.0
*p*-Coumaric acid	1.2 ± 0.0	ND
Caffeic acid	ND	0.1 ± 0.0
Synaptic acid	0.6 ± 0.0	0.9 ± 0.1
Sinapaldehyde	ND	1.0 ± 0.0
*Flavonoids*		
(−)-Epicatechin	833.1 ± 2.7	81.5 ± 3.4
(+)-Catechin	573.8 ± 11.2	88.8 ± 3.4
Epigallocatechin-3-gallate	785.0 ± 1.6	8.4 ± 0.1
Quercetin	231.7 ± 1.2	15.2 ± 0.2
Quercetin-3-rutinoside	72.6 ± 1.0	13.1 ± 0.6
Quercetin-3-glucoside	2.5 ± 0.1	35.3 ± 1.6
Hesperidin	9.1 ± 0.7	12.7 ± 0.3
Kaempferol	ND	1.8 ± 0.1
Kaempferol-3-rutinoside	1.5 ± 0.1	ND
Taxifolin	0.4 ± 0.1	ND
Pinocembrine	ND	1.0 ± 0.1
Galangine	ND	0.1 ± 0.0
Apigenin	ND	0.2 ± 0.0
*Others*		
Coniferaldehyde	ND	0.5 ± 0.1
Total identified (mg/100 g)	2519	329

Note: ND = not detected.

## Data Availability

Not available.

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
