# Peer review of "Optimization of the Green Chemistry-like Extraction of Phenolic Compounds from Grape (Vitis labrusca L.) and Blackberry (Rubus fruticosus L.) Seeds with Concomitant Biological and Antioxidant Activity Assessments"

_plants, 2023, doi:10.3390/plants12142618_

Round 1
Reviewer 1 Report (Previous Reviewer 1)
The manuscripts looks much better. It seems that auhtors have taken seriosly the comments given before. Still there are many typhos errors, including many instances with the extra spaces.
English still needs to be polished. Several sentences are too long. Suggestion is to look for professional English editing service. MDPI has the very good and very affordable service, among others. The pther optio nis to purchase Grammarly Premium or Instatext Editor programmes.
Author Response
Thanks for the suggestion. The initial submission was revised by a certified English translator and the revision was corrected using a different service. Please check the changes in the revised version.
Reviewer 2 Report (Previous Reviewer 2)
I read this article with interest that aimed to provide an update on the antioxidant activity of polyphenols, their isolation, and identification. Much work has been done, but the submitted manuscript should be revised. I encourage the authors to consider making the necessary changes.
"Answer: All changes were made in the text." (Response to Reviewer 2 Comments)
This statement is not entirely correct. The authors did not calculate the correlation coefficient and change it. Also, the authors did not specify the Uv-VIS spectrophotometer on which the measurement was performed.
Materials and Methods
Please specify the Uv-VIS spectrophotometer on which the measurement was performed. Phenolic composition by UV-VIS spectrophotometry and Antioxidant activity are insufficiently described.
- Line 156 (plants-2447185 version 1) and line 169 (plants-2494263 version): According to the ICH guidelines, the linearity requirement is a correlation coefficient (r)> 0.999, not a determination coefficient (R2). Therefore, the authors should calculate the correlation coefficient and change it.
Unfortunately, many mistakes affect the reading and understanding of this manuscript. Several typing errors should be corrected.
- Lines 76, 104: "in vitro "
- Page 12/34: "60% ethanol "
Please, check the manuscript carefully, and unify the presentation.
Based on the above criticism, I recommend minor manuscript revisions before resubmitting.
Therefore, I recommend the publication of the submitted manuscript in the Journal after minor revision.
The English should be improved throughout the text, and the typing errors should be corrected.
Author Response
R: Okay, the UV-VIS spectrophotometer has been specified. Phenolic composition by UV-VIS spectrophotometry and antioxidant activity were described in more detail. Thanks!
R: Ok, the correlation coefficient has been calculated and changed. Thanks!
R: Okay, the manuscript has been carefully checked and corrected. Thanks!
Reviewer 3 Report (Previous Reviewer 4)
Now, after all corrections paper looks far better and could be published as it is.
Author Response
Thanks for your initial comments and suggestions.
This manuscript is a resubmission of an earlier submission. The following is a list of the peer review reports and author responses from that submission.
Round 1
Reviewer 1 Report
General comment
The work of Tufy Kabbas Junior, Cristiane de Moura, Thiago Mendanha Cruz, Mariza Boscacci Marques, Mariana Araújo Vieira do Carmo, Carolina Turnes Pasini Deolindo, Heitor Daguer, Luciana Azevedo and Daniel Granato, entitled “Grape (Vitis labrusca cv. Bordeaux) and blackberry (Rubus fruticosus) seeds: Optimising the extraction of phenolic compounds with cellular antioxidant activity” has many of experimental works, but the novelty is lacking or at least it is not brought up by adequate and responsible scientific writing. It is not clear what is the specifity and novelty of extraction method employed, if any. There are many instances in the text with the exaggerations, informal language and incorrect, superficial statements. This study requires de novo writing since it is very chaotic, many aspects were not covered, explained, outlined. There is a complete mess with Tables, their mentioning in the text and some supplementary material and appendices, which should all be placed in one file of Supplementary material. The graphical work is of unaccepted quality, very small letters and poor resolution, preventing a fair review. The English language grammar and style require substantial improvement. At the end as summary, what is really a novel piece of information here? Perhaps, authors should look for another more suitable journal to place their work.
Please see some examples of minor specific comments down below (this is not a definite list, just examples).
1. Title: instead of the present title, consider replacing it with:
“Optimization of the green chemistry- like extraction of phenolic compounds from grape (Vitis labrusca L.) and blackberry (Rubus fruticosus L.) seeds with concomitant biological and antioxidant activity assessment”
2. What about EA.hy926 (endothelial cell hybrid line, it was not mentioned much in the results and also not in abstract. Special considerations should be paid to results done on this cell lien since it is not cancer genic and immortalized cell line. I see differences between the cancer immortalized cell lines and this one, which have not been addressed at all.
3. Abstract: A wording” Soxhlet (Sox)” has to be amended into Soxhlet extraction or Soxhlet apparatus. There are many of such examples throughout the text that point to informal and inappropriate expressions.
4. Introduction (example of unclear and inaccurate presentation of published work, possibly due to lack of skills in the English language understanding and expression)
In the text: “In wine production, large amounts of waste are generated, around 20% of the total weight (grape seeds), which are increasingly used to recover essential oils, phenolic compounds, and fibers, becoming a sustainable waste management strategy. “
This sentence conveys to a reader that there is approximately 20% of waste from the grape seed in vine industry, which is not the case. Please see the reference, specifically this sentence:” After the wine production, a considerable amount of solid wastes namely grape pomace or grape marc and lees are generated. Solids containing skins, seeds, and stems are regarded as grape marc or grape pomace, and represent approximately 20% of the total processed grapes” from https://doi.org/10.1016/j.cogsc.2020.100415
Authors should write 20% of grape berries instead of grape seeds. One berry of grape contains skin, seeds, pulp, juice…
The English language grammar and style require substantial improvement.It needs assistance of the professional service with the background in the life sciences.
Reviewer 2 Report
The manuscript presents interesting and valuable work within the scope of the Plants. Lots of work has been done. However, from a scientific point of view, the manuscript is well-organized and needs no substantial changes.
Some elements of experimental analysis, interpretation, and presentation require corrections. Several considerations should be taken into account before publication. Below you can find a list of my comments and suggestions:
Materials and Methods
Please specify the Uv-VIS spectrophotometer on which the measurement was performed. Phenolic composition by UV-VIS spectrophotometry and Antioxidant activity are insufficiently described.
- Line 156: According to the ICH guidelines, the linearity requirement is a correlation coefficient (r)> 0.999, not a determination coefficient (R2). Therefore, the authors should calculate the correlation coefficient and change it.
Unfortunately, many mistakes affect the reading and understanding of this manuscript. Several typing errors should be corrected
- Line 45: „[3,4,5,6]“
- Line 77: „Reagent“, „Sodium Hydroxide“
- Line 78: „3.3 ' acid, 4-78 trihydroxyfuchsone-2'-sulfonic acid“
- Line 223: „Ethanol“
Please, check the manuscript carefully, and unify the presentation.
The interpretation of obtained results is far below the Journal's quality standards.
In my opinion, the proposed manuscript is acceptable as the publication suitable for Plants after major revision.
The English should be improved throughout the text and the typing errors should be corrected.
Reviewer 3 Report
The manuscript submitted to me for review was titled “Grape (Vitis labrusca cv. Bordeaux) and blackberry (Rubus fruticosus) seeds: Optimising the extraction of phenolic compounds with cellular antioxidant activity” The authors mentioned a great idea to employ biorefinery concept, where each single component of the production chain is considered as a potential valuable product. Experiments were planned with this idea in mind. The aim of the study was to optimize extraction parameters of bioactive compounds from grape (Vitis labrusca) and blackberry (Rubus fruticosus) seeds. The authors evaluated the composition, antioxidant activity, and cytotoxicity of the extracts. The great value of this research is the verification not only of the content of antioxidants, including polyphenols in the obtained extracts, but also their cytotoxicity and antihemolytic activity in vitro. Results indicate that the extracts obtained from grape and blackberry seeds possess significant bioactivity, exhibit a protective effect against ROS production induced by H2O2. They also demonstrated antioxidant activity and provided protection to erythrocytes when subjected to hypotonic and isotonic conditions, without exhibiting any hemolytic activity. The discussion of the results is interesting, but the presentation of the results can be improved. Figures in particular should be place close to the text in which they are described. My overall assessment of the manuscript is positive, I believe that with minor corrections it can be published.
Reviewer 4 Report
This paper describes the composition of antioxidant phenolics isolated from seesd of Brazilian strains of grape ane blackberry. It is a typical work on biorefining of food industrywaste. Thus, it is of interest tofood scientists. Similar studies have been, however, carried out for grape previously using other extrahent (methanol versus ethanol) Thus this part is of smaller novelty (despite that Authors put stress on ethanol as green solvent). Seeds were previously "degreased" by extraction with hexane in Soxhlet apparatus.
Paper could be published, however, requires serious rewriting. My suggestions are as follows:
0/ paper is generally written quite chaotically, Authors please rethink the set up of the paper; In contradiction Experimental Part is written very well;.
1./ degreasing procedure results are completely omitted in the paper - what was the product? was its composition being studied? (oils obtained from such ususual sources are often very interesting compounds), or at least what is it content of oil in the seeds.
2./ Authors overdiscuss the use of algorithms used for optimization of the extraction procedure (final one - with ethanol). Such optimization is quite populalry used now. Results should be described in the paper shortly, when tables should be moved to Supplementary Part. The use of this system provides final procedure of extraction, however the error analysis should be made before stating that for 1 g of seeds 61.29 mL of solvent is required (this accuracy is irrational);
3./Typical chromatogram(s) should be presented in the paper - most likely as Supplementary part;
3./names of chemical compounds should be wtitten in small letters (especially etanol);
4./ Abbreviations should be explained when used for the first time in the text;
English is at the lowest acceptance level, however its improvement would be beneficial for the paper
Reviewer 5 Report
Dear Author,
Kindly response to the below comments:
Please specify which varieties you used in your study for grape and blackberry (add photos if possible and a short description for both plants).
Abstract: Kindly add more details in this section especially on the phytochemical composition of used seed extracts.
Authors have written in the Abstract: Simultaneous optimization was applied with a combination of 35.9 min of extraction and a solid-to-solvent ratio of 1 g defatted grape seed to 61.28 mL extracting solvent (Ethanol 60%) and 62.1 min of extraction and a solid-to-solvent ratio of 1 g defatted blackberry seed to 64.1 mL extracting solvent (60% ethanol): I can’t find these combinations (extraction time/solvent volume) in the Table 1 and 2? Kindly give more explanation.
Line 25: kindly correct: H2O2
Line 56: kindly correct the typo
Kindly give explanation (more details) about the choice of both grape seeds and blackberry seeds
Introduction: it’s very short, kindly add more details on the used seed extracts (chemical compositions,….) and describe the obtained result of other authors.
The figure and table legends need improvement. All legends should have enough description for a reader to understand the figure/table without having to refer back to the main text of the manuscript.
Result and discussion section: 7 tables (a lot of information in each table) and 3 figures (in total 16 graphs): the text is not proportional to the mass of provided results, for that I suggest to the authors to develop more the text in this section. On the other hand, the material and methods section is very well detailed, which makes the paper not balanced/ homogeneous.
Material and Methods:
Line 136 and 137: kindly move this sentence to result and discussion section, also in Table 7, I counted only 22 phenolic compounds and not 35 as you mention in the text: Line 136 (grape 13, blackberry: 19)???
Kindly add the obtained chromatograms in supplementary data
Minor editing of English language required